# Understanding Composition of Word Embeddings via Tensor Decomposition

**Abraham Frandsen & Rong Ge**
Department of Computer Science
Duke University
Durham, NC 27708, USA
`{abef,rongge}@cs.duke.edu`

## Abstract

Word embedding is a powerful tool in natural language processing. In this paper we consider the problem of word embedding composition – given vector representations of two words, compute a vector for the entire phrase. We give a generative model that can capture specific syntactic relations between words. Under our model, we prove that the correlations between three words (measured by their PMI) form a tensor that has an approximate low rank Tucker decomposition. The result of the Tucker decomposition gives the word embeddings as well as a core tensor, which can be used to produce better compositions of the word embeddings. We also complement our theoretical results with experiments that verify our assumptions, and demonstrate the effectiveness of the new composition method.

## 1 Introduction

Word embeddings have become one of the most popular techniques in natural language processing. A word embedding maps each word in the vocabulary to a low dimensional vector. Several algorithms (e.g., Mikolov et al. (2013); Pennington et al. (2014)) can produce word embedding vectors whose distances or inner-products capture semantic relationships between words. The vector representations are useful for solving many NLP tasks, such as analogy tasks(Mikolov et al., 2013) or serving as features for supervised learning problems (Maas et al., 2011).

While word embeddings are good at capturing the semantic information of a single word, a key challenge is the problem of *composition*: how to combine the embeddings of two co-occurring, syntactically related words to an embedding of the entire phrase. In practice composition is often done by simply adding the embeddings of the two words, but this may not be appropriate when the combined meaning of the two words differ significantly from the meaning of individual words (e.g., "complex number" should not just be "complex"+"number").

In this paper, we try to learn a model for word embeddings that incorporates syntactic information and naturally leads to better compositions for syntactically related word pairs. Our model is motivated by the principled approach for understanding word embeddings initiated by Arora et al. (2015), and models for composition similar to Coecke et al. (2010).

Arora et al. (2015) gave a generative model (RAND-WALK) for word embeddings, and showed several previous algorithms can be interpreted as finding the hidden parameters of this model. However, the RAND-WALK model does not treat syntactically related word-pairs differently from other word pairs. We give a generative model called syntactic RAND-WALK (see Section 3) that is capable of capturing specific syntactic relations (e.g., adjective-noun or verb-object pairs). Taking adjective-noun pairs as an example, previous works (Socher et al., 2012; Baroni & Zamparelli, 2010; Maillard & Clark, 2015) have tried to model the adjective as a linear operator (a matrix) that can act on the embedding of the noun. However, this would require learning a $d \times d$ matrix for each adjective while the normal embedding only has dimension $d$. In our model, we use a core tensor $T \in \mathbb{R}^{d \times d \times d}$ to capture the relations between a pair of words and its context. In particular, using the tensor $T$ and the word embedding for the adjective, it is possible to define a matrix for the adjective that can be used as an operator on the embedding of the noun. Therefore our model allows the same interpretations as many previous models while having much fewer parameters to train.

One salient feature of our model is that it makes good use of high order statistics. Standard word embeddings are based on the observation that the semantic information of a word can be captured by words that appear close to it. Hence most algorithms use pairwise co-occurrence between words to learn the embeddings. However, for the composition problem, the phrase of interest already has two words, so it would be natural to consider co-occurrences between at least three words (the two words in the phrase and their neighbors).

Based on the model, we can prove an elegant relationship between high order co-occurrences of words and the model parameters. In particular, we show that if we measure the Pointwise Mutual Information (PMI) between three words, and form an $n \times n \times n$ tensor that is indexed by three words $a, b, w$, then the tensor has a Tucker decomposition that exactly matches our core tensor $T$ and the word embeddings (see Section 2, Theorem 1, and Corollary 1). This suggests a natural way of learning our model using a tensor decomposition algorithm.

Our model also allows us to approach the composition problem with more theoretical insights. Based on our model, if words $a, b$ have the particular syntactic relationships we are modeling, their composition will be a vector $v_a + v_b + T(v_a, v_b, \cdot)$. Here $v_a, v_b$ are the embeddings for word $a$ and $b$, and the tensor gives an additional correction term. By choosing different core tensors it is possible to recover many previous composition methods. We discuss this further in Section 3.

Finally, we train our new model on a large corpus and give experimental evaluations. In the experiments, we show that the model learned satisfies the new assumptions that we need. We also give both qualitative and quantitative results for the new embeddings. Our embeddings and the novel composition method can capture the specific meaning of adjective-noun phrases in a way that is impossible by simply "adding" the meaning of the individual words. Quantitative experiment also shows that our composition vector are better correlated with humans on a phrase similarity task.

## 1.1 RELATED WORK

**Syntax and word embeddings**  Many well-known word embedding methods (e.g., Pennington et al. (2014); Mikolov et al. (2013)) don't explicitly utilize or model syntactic structure within text. Andreas & Klein (2014) find that such syntax-blind word embeddings fail to capture syntactic information above and beyond what a statistical parser can obtain, suggesting that more work is required to build syntax into word embeddings.

Several syntax-aware embedding algorithms have been proposed to address this. Levy & Goldberg (2014a) propose a syntax-oriented variant of the well-known skip-gram algorithm of Mikolov et al. (2013), using contexts generated from syntactic dependency-based contexts obtained with a parser. Cheng & Kartsaklis (2015) build syntax-awareness into a neural network model for word embeddings by introducing a negative set of samples in which the order of the context words is shuffled, in hopes that the syntactic elements which are sensitive to word order will be captured.

**Word embedding composition**  Several works have addressed the problem of composition for word embeddings. On the theoretical side, Gittens et al. (2017) give a theoretical justification for additive embedding composition in word models that satisfy certain assumptions, such as the skip-gram model, but these assumptions don't address syntax explicitly. Coecke et al. (2010) present a mathematical framework for reasoning about syntax-aware word embedding composition that motivated our syntactic RAND-WALK model. Our new contribution is a concrete and practical learning algorithm with theoretical guarantees. Mitchell & Lapata (2008; 2010) explore various composition methods that involve both additive and multiplicative interactions between the component embeddings, but some of these are limited by the need to learn additional parameters post-hoc in a supervised fashion.

Guevara (2010) get around this drawback by first training word embeddings for each word and also for tokenized adjective-noun pairs. Then, the composition model is trained by using the constituent adjective and noun embeddings as input and the adjective-noun token embedding as the predictive target. Maillard & Clark (2015) treat adjectives as matrices and nouns as vectors, so that the composition of an adjective and noun is just matrix-vector multiplication. The matrices and vectors are learned through an extension of the skip-gram model with negative sampling. In contrast to these approaches, our model gives rise to a syntax-aware composition function, which can be learned

along with the word embeddings in an unsupervised fashion, and which generalizes many previous composition methods (see Section 3.3 for more discussion).

**Tensor factorization for word embeddings**   As Levy & Goldberg (2014b) and Li et al. (2015) point out, some popular word embedding methods are closely connected matrix factorization problems involving pointwise mutual information (PMI) and word-word co-occurrences. It is natural to consider generalizing this basic approach to tensor decomposition. Sharan & Valiant (2017) demonstrate this technique by performing a CP decomposition on triple word co-occurrence counts. Bailey & Aeron (2017) explore this idea further by defining a third-order generalization of PMI, and then performing a symmetric CP decomposition on the resulting tensor. In contrast to these recent works, our approach arives naturally at the more general Tucker decomposition due to the syntactic structure in our model. Our model also suggests a different (yet still common) definition of third-order PMI.

## 2   PRELIMINARIES

**Notation**   For a vector $v$, we use $\|v\|$ to denote its Euclidean norm. For vectors $u, v$ we use $\langle u, v \rangle$ to denote their inner-product. For a matrix $M$, we use $\|M\|$ to denote its spectral norm, $\|M\|_F = \sqrt{\sum_{i,j} M_{i,j}^2}$ to denote its Frobenius norm, and $M_{i,:}$ to denote it's $i$-th row. In this paper, we will also often deal with 3rd order tensors, which are just three-way indexed arrays. We use $\otimes$ to denote the tensor product: if $u, v, w \in \mathbb{R}^d$ are $d$-dimensional vectors, $T = u \otimes v \otimes w$ is a $d \times d \times d$ tensor whose entries are $T_{i,j,k} = u_i v_j w_k$.

**Tensor basics**   Just as matrices are often viewed as bilinear functions, third order tensors can be interpreted as trilinear functions over three vectors. Concretely, let $T$ be a $d \times d \times d$ tensor, and let $x, y, z \in \mathbb{R}^d$. We define the scalar $T(x, y, z) \in \mathbb{R}$ as follows

$$T(x, y, z) = \sum_{i,j,k=1}^{d} T_{i,j,k} x(i) y(j) z(k).$$

This operation is linear in $x, y$ and $z$. Analogous to applying a matrix $M$ to a vector $v$ (with the result vector $Mv$), we can also apply a tensor $T$ to one or two vectors, resulting in a matrix and a vector, respectively:

$$T(x, y, \cdot)(k) = \sum_{i,j=1}^{d} T_{i,j,k} x(i) y(j), \quad T(x, \cdot, \cdot)_{j,k} = \sum_{i=1}^{d} T_{i,j,k} x(i)$$

We will make use of the simple facts that $\langle z, T(x, y, \cdot) \rangle = T(x, y, z)$ and $[T(x, \cdot, \cdot)]^\top y = T(x, y, \cdot)$.

**Tensor decompositions**   Unlike matrices, there are several different definitions for the *rank* of a tensor. In this paper we mostly use the notion of *Tucker rank* Tucker (1966). A tensor $T \in \mathbb{R}^{n \times n \times n}$ has Tucker rank $d$, if there exists a core tensor $S \in \mathbb{R}^{d \times d \times d}$ and matrices $A, B, C \in \mathbb{R}^{n \times d}$ such that

$$T_{i,j,k} = \sum_{i',j',k'=1}^{d} S_{i',j',k'} A_{i,i'} B_{j,j'} C_{k,k'} = S(A_{i,:}, B_{j,:}, C_{k,:}),$$

The equation above is also called a *Tucker decomposition* of the tensor $T$. The Tucker decomposition for a tensor can be computed efficiently.

When the core tensor $S$ is restricted to a diagonal tensor (only nonzero at entries $S_{i,i,i}$), the decomposition is called a CP decomposition Carroll & Chang (1970); Harshman (1970) which can also be written as $T = \sum_{i=1}^{d} S_{i,i,i} A_{i,:} \otimes B_{i,:} \otimes C_{i,:}$. In this case, the tensor $T$ is the sum of $d$ rank-1 tensors $(A_{i,:} \otimes B_{i,:} \otimes C_{i,:})$. However, unlike matrix factorizations and the Tucker decomposition, the CP decomposition of a tensor is hard to compute in the general case (Håstad, 1990; Hillar & Lim, 2013). Later in Section 4 we will also see why our model for syntactic word embeddings naturally leads to a Tucker decomposition.

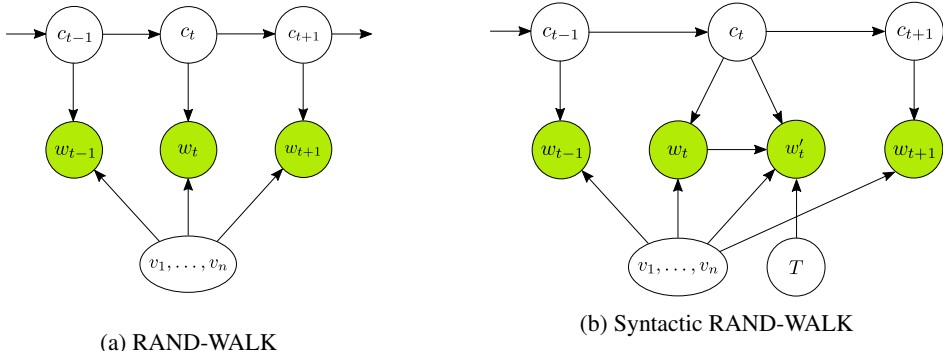

(a) RAND-WALK        (b) Syntactic RAND-WALK

Figure 1: Graphical models of RAND-WALK (left) and our new model (right), depicting a syntactic word pair $(w_t, w_t')$. Green nodes correspond to observed variables, white nodes to latent variables.

## 3   SYNTACTIC RAND-WALK MODEL

In this section, we introduce our syntactic RAND-WALK model and present formulas for inference in the model. We also derive a novel composition technique that emerges from the model.

**RAND-WALK model**   We first briefly review the RAND-WALK model (Arora et al., 2015). In this model, a corpus of text is considered as a sequence of random variables $w_1, w_2, w_3, \ldots$, where $w_t$ takes values in a vocabulary $V$ of $n$ words. Each word $w \in V$ has a word embedding $v_w \in \mathbb{R}^d$. The prior for the word embeddings is $v_w = s \cdot \hat{v}$, where $s$ is a positive bounded scalar random variable with constant expectation $\tau$ and upper bound $\kappa$, and $\hat{v} \sim N(0, I)$.

The distribution of each $w_t$ is determined in part by a random walk $\{c_t \in \mathbb{R}^d \,|\, t = 1, 2, 3 \ldots\}$, where $c_t$ – called a *discourse vector* – represents the topic of the text at position $t$. This random walk is slow-moving in the sense that $\|c_{t+1} - c_t\|$ is small, but mixes quickly to a stationary distribution that is uniform on the unit sphere, which we denote by $\mathcal{C}$.

Let $\mathscr{C}$ denote the sequence of discourse vectors, and let $\mathscr{V}$ denote the set of word embeddings. Given these latent variables, the model specifies the following conditional probability distribution:

$$\Pr[w_t = w \,|\, c_t] \propto \exp(\langle v_w, c_t \rangle). \tag{1}$$

The graphical model depiction of RAND-WALK is shown in Figure 1a.

### 3.1   SYNTACTIC RAND-WALK

One limitation of RAND-WALK is that it can't deal with syntactic relationships between words. Observe that conditioned on $c_t$ and $\mathscr{V}$, $w_t$ is independent of the other words in the text. However, in natural language, words can exhibit more complex dependencies, e.g. adjective-noun pairs, subject-verb-object triples, and other syntactic or grammatical structures.

In our syntactic RAND-WALK model, we start to address this issue by introducing direct pairwise word dependencies in the model. When there is a direct dependence between two words, we call the two words a *syntactic word pair*. In RAND-WALK, the interaction between a word embedding $v$ and a discourse vector $c$ is mediated by their inner product $\langle v, c \rangle$. When modeling a syntactic word pair, we need to mediate the interaction between *three* quantities, namely a discourse vector $c$ and the word embeddings $v$ and $v'$ of the two relevant words. A natural generalization is to use a trilinear form defined by a tensor $T$, i.e.

$$T(v, v', c) = \sum_{i,j,k=1}^{d} T_{i,j,k} v(i) v'(j) c(k).$$

Here, $T \in \mathbb{R}^{d \times d \times d}$ is also a latent random variable, which we call the *composition tensor*.

We model a syntactic word pair as a single semantic unit within the text (e.g. in the case of adjective-noun phrases). We realize this choice by allowing each discourse vector $c_t$ to generate a pair of words $w_t, w'_t$ with some small probability $p_{syn}$. To generate a syntactic word pair $w_t, w'_t$, we first generate a *root word* $w_t$ conditioned on $c_t$ with probability proportional to $\exp(\langle c_t, w_t \rangle)$, and then we draw $w'_t$ from a conditional distribution defined as follows:

$$\Pr[w'_t = b \,|\, w_t = a, \mathscr{C}, \mathscr{V}] \propto \exp(\langle c_t, v_b \rangle + T(v_a, v_b, c_t)). \tag{2}$$

Here $\exp(\langle c_t, v_b \rangle)$ would be proportional to the probability of generating word $b$ in the original RAND-WALK model, without considering the syntactic relationship. The additional term $T(v_a, v_b, c_t)$ can be viewed as an adjustment based on the syntactic relationship.

We call this extended model Syntactic RAND-WALK. Figure 1b gives the graphical model depiction for a syntactic word pair, and we summarize the model below.

**Definition 1** (Syntactic RAND-WALK model). *The model consists of the following:*

1. *Each word $w$ in vocabulary has a corresponding embedding $v_w \sim s \cdot \hat{v}_w$, where $s \in \mathbb{R}_{\geq 0}$ is bounded by $\kappa$ and $\mathbb{E}[s] = \tau$; $\hat{v}_w \sim N(0, I_{d \times d})$.*

2. *The sequence of discourse vectors $c_1, ..., c_t$ are generated by a random walk on the unit sphere, $\|c_t - c_{t+1}\| \leq \epsilon_w / \sqrt{d}$ and the stationary distribution is uniform.*

3. *For each $c_t$, with probability $1 - p_{syn}$, it generates one word $w_t$ with probability proportional to $\exp(\langle c_t, v_{w_t} \rangle)$.*

4. *For each $c_t$, with probability $p_{syn}$, it generates a syntactic pair $w_t, w'_t$ with probability proportional to $\exp(\langle c_t, v_{w_t} \rangle)$ and $\exp(\langle c_t, v_{w'_t} \rangle + T(v_{w_t}, v_{w'_t}, c_t))$ respectively, where $T$ is a $d \times d \times d$ composition tensor.*

### 3.2 INFERENCE IN THE MODEL

We now calculate the marginal probabilities of observing pairs and triples of words under the syntactic RAND-WALK model. We will show that these marginal probabilities are closely related to the model parameters (word embeddings and the composition tensor). All proofs in this section are deferred to supplementary material.

Throughout this section, we consider two adjacent context vectors $c_t$ and $c_{t+1}$, and condition on the event that $c_t$ generated a single word and $c_{t+1}$ generated a syntactic pair[1]. The main bottleneck in computing the marginal probabilities is that the conditional probabilities specified in equations (1) and (2) are not normalized. Indeed, for these equations to be exact, we would need to divide by the appropriate partition functions, namely $Z_{c_t} := \sum_{w \in V} \exp(\langle v_w, c_t \rangle)$ for the former and $Z_{c_t, a} := \sum_{w \in V} \exp(\langle c_t, v_w \rangle + T(v_a, v_w, c_t))$ for the latter. Fortunately, we show that under mild assumptions these quantities are highly concentrated. To do that we need to control the norm of the composition tensor.

**Definition 2.** *The composition tensor $T$ is $(K, \epsilon)$-bounded, if for any word embedding $v_a, v_b$, we have*

$$\|T(v_a, \cdot, \cdot) + I\|^2 \leq \frac{Kd\epsilon^2}{\log^2 n}; \quad \|T(v_a, \cdot, \cdot) + I\|_F^2 \leq Kd; \quad \|T(v_a, v_b, \cdot)\|^2 \leq Kd.$$

To make sure $\exp(\langle c_t, v_w \rangle + T(v_a, v_w, c_t))$ are within reasonable ranges, the value $K$ in this definition should be interpreted as an absolute constant (like 5, similar to previous constants $\kappa$ and $\tau$). Intuitively these conditions make sure that the effect of the tensor cannot be too large, while still making sure the tensor component $T(v_a, v_b, c)$ can be comparable (or even larger than) $\langle v_b, c \rangle$. We have not tried to optimize the log factors in the constraint for $\|T(v_a, \cdot, \cdot) + I\|^2$.

Note that if the tensor component $T(v_a, \cdot, \cdot)$ has constant singular values (hence comparable to $I$), we know these conditions will be satisfied with $K = O(1)$ and $\epsilon = O(\frac{\log n}{\sqrt{d}})$. Later in Section 5 we verify that the tensors we learned indeed satisfy this condition. Now we are ready to state the concentration of partition functions:

---

[1] As we will see in Section 5, in practice it is easy to identify which words form a syntactic pair, so it is possible to condition on this event in training.

**Lemma 1** (Concentration of partition functions). *For the syntactic RAND-WALK model, there exists a constant $Z$ such that*

$$\Pr_{c \sim \mathcal{C}}[(1 - \epsilon_z)Z \le Z_c \le (1 + \epsilon_z)Z] \ge 1 - \delta,$$

*for $\epsilon_z = \tilde{O}(1/\sqrt{n})$ and $\delta = \exp(-\Omega(\log^2 n))$.*

*Furthermore, if the tensor $T$ is $(K, \epsilon)$-bounded, then for any fixed word $a \in V$, there exists a constant $Z_a$ such that*

$$\Pr_{c \sim \mathcal{C}}[(1 - \epsilon_{z,a})Z_a \le Z_{c,a} \le (1 + \epsilon_{z,a})Z_a] \ge 1 - \delta,$$

*for $\epsilon_{z,a} = O(\epsilon) + \tilde{O}(1/\sqrt{n})$ and $\delta = \exp(-\Omega(\log^2 n))$.*

Using this lemma, we can obtain simple expressions for co-occurrence probabilities. In particular, for any fixed $w, a, b \in V$, we adopt the following notation:

$$p(a) := \Pr[w_{t+1} = a] \quad p(w, a) := \Pr[w_t = w, w_{t+1} = a]$$

$$p([a, b]) := \Pr[w_{t+1} = a, w'_{t+1} = b] \quad p(w, [a, b]) := \Pr[w_t = w, w_{t+1} = a, w'_{t+1} = b].$$

Here in particular we use $[a, b]$ to highlight the fact that $a$ and $b$ form a syntactic pair. Note $p(w, a)$ is the same as the co-occurrence probability of words $w$ and $a$ if both of them are the only word generated by the discourse vector. Later we will also use $p(w, b)$ to denote $\Pr[w_t = w, w_{t+1} = b]$ (not $\Pr[w_t = w, w'_{t+1} = b]$).

We also require two additional properties of the word embeddings, namely that they are norm-bounded above by some constant times $\sqrt{d}$, and that all partition functions are bounded below by a positive constant. Both of these properties hold with high probability over the word embeddings provided $n \gg d \log d$ and $d \gg \log n$, as shown in the following lemma:

**Lemma 2.** *Assume that the composition tensor $T$ is $(K, \epsilon)$-bounded, where $K$ is a constant. With probability at least $1 - \delta_1 - \delta_2$ over the word vectors, where $\delta_1 = \exp(\Theta(d \log d) - \Theta(n))$ and $\delta_2 = \exp(\Theta(\log n) - \Theta(d))$, there exist positive absolute constants $\gamma$ and $\beta$ such that $\|v_i\| \le \kappa\gamma$ for each $i \in V$ and $Z_c \ge \beta$ and $Z_{c,a} \ge \beta$ for any unit vector $c \in \mathbb{R}^d$ and any word $a \in V$.*

We can now state the main result.

**Theorem 1.** *Suppose that the events referred to in Lemma 1 hold. Then*

$$\log p(a) = \frac{\|v_a\|^2}{2d} - \log Z \pm \epsilon_p \tag{3}$$

$$\log p(w, a) = \frac{\|v_w + v_a\|^2}{2d} - 2\log Z \pm \epsilon_p \tag{4}$$

$$\log p([a, b]) = \frac{\|v_a + v_b + T(v_a, v_b, \cdot)\|^2}{2d} - \log Z - \log Z_a \pm \epsilon_p \tag{5}$$

$$\log p(w, [a, b]) = \frac{\|v_w + v_a + v_b + T(v_a, v_b, \cdot)\|^2}{2d} - 2\log Z - \log Z_a \pm \epsilon_p \tag{6}$$

*Here $\epsilon_p = O(\epsilon + \epsilon_w) + \tilde{O}(1/\sqrt{n} + 1/d)$, where $\epsilon$ is from the $(K, \epsilon)$-boundedness of $T$ and $\epsilon_w$ is from Definition 1.*

### 3.3 COMPOSITION

Our model suggests that the latent discourse vectors contain the meaning of the text at each location. It is therefore reasonable to view the discourse vector $c$ corresponding to a syntactic word pair $(a, b)$ as a suitable representation for the phrase as a whole. The posterior distribution of $c$ given $(a, b)$ satisfies

$$\Pr[c_t = c \mid w_t = a, w'_t = b] \propto \frac{1}{Z_c Z_{c,a}} \exp\left(\langle v_a + v_b + T(v_a, v_b, \cdot), c \rangle\right) \Pr[c_t = c].$$

Since $\Pr[c_t = c]$ is constant, and since $Z_c$ and $Z_{c,a}$ concentrate on values that don't depend on $c$, the MAP estimate of $c$ given $[a, b]$, which we denote by $\hat{c}$, satisfies

$$\hat{c} \approx \arg\max_{\|c\|=1} \exp\left(\langle v_a + v_b + T(v_a, v_b, \cdot), c \rangle\right) = \frac{v_a + v_b + T(v_a, v_b, \cdot)}{\|v_a + v_b + T(v_a, v_b, \cdot)\|}.$$

Hence, we arrive at our basic tensor composition: for a syntactic word pair $(a, b)$, the composite embedding for the phrase is $v_a + v_b + T(v_a, v_b, \cdot)$.

Note that our composition involves the traditional additive composition $v_a + v_b$, plus a correction term $T(v_a, v_b, \cdot)$. We can view $T(v_a, v_b, \cdot)$ as a matrix-vector multiplication $[T(v_a, \cdot, \cdot)]^\top v_b$, i.e. the composition tensor allows us to compactly associate a matrix with each word in the same vein as Maillard & Clark (2015). Depending on the actual value of $T$, the term $T(v_a, v_b, \cdot)$ can also recover any manner of linear or multiplicative interactions between $v_a$ and $v_b$, such as those proposed in Mitchell & Lapata (2010).

## 4 LEARNING

In this section we discuss how to learn the parameters of the syntactic RAND-WALK model. Theorem 1 provides key insights into the learning problem, since it relates joint probabilities between words (which can be estimated via co-occurrence counts) to the word embeddings and composition tensor. By examining these equations, we can derive a particularly simple formula that captures these relationships. To state this equation, we define the PMI for 3 words as

$$PMI3(a, b, w) := \log \frac{p(w, [a, b])p(a)p(b)p(w)}{p(w, a)p(w, b)p([a, b])}. \tag{7}$$

We note that this is just one possible generalization of pointwise mutual information (PMI) to several random variables, but in the context of our model, it is a very natural definition as all the partition numbers will be canceled out. Indeed, as an immediate corollary of Theorem 1, we have

**Corollary 1.** *Suppose that the events referred to in Lemma 1 hold. Then for $\epsilon_p$ same as Theorem 1*

$$PMI3(a, b, w) = \frac{1}{d}T(v_a, v_b, v_w) \pm O(\epsilon_p). \tag{8}$$

That is, if we consider $PMI3(a, b, w)$ as a $n \times n \times n$ tensor, Equation equation 8 is exactly a Tucker decomposition of this tensor of Tucker rank $d$. Therefore, all the parameters of the syntactic RAND-WALK model can be obtained by finding the Tucker decomposition of the PMI3 tensor. This equation also provides a theoretical motivation for using third-order pointwise mutual information in learning word embeddings.

### 4.1 IMPLEMENTATION

We now discuss concrete details about our implementation of the learning algorithm.[2]

**Corpus.** We train our model using a February 2018 dump of the English Wikipedia. The text is pre-processed to remove non-textual elements, stopwords, and rare words (words that appear less than 1000 within the corpus), resulting in a vocabulary of size 68,279. We generate a matrix of word-word co-occurrence counts using a window size of 5. To generate the tensors of adjective-noun-word and verb-object-word co-occurrence counts, we first run the Stanford Dependency Parser (Chen & Manning, 2014) on the corpus in order to identify all adjective-noun and verb-object word pairs, and then use context windows that don't cross sentence boundaries to populate the triple co-occurrence counts.

**Training.** We first train the word embeddings according to the RAND-WALK model, following Arora et al. (2015). Using the learned word embeddings, we next train the composition tensor $T$ via the following optimization problem

$$\min_{T, \{C_w\}, C} \sum_{(a,b),w} f(X_{(a,b),w}) \left(\log(X_{(a,b),w}) - \|v_w + v_a + v_b + T(v_a, v_b, \cdot)\|^2 - C_a - C\right)^2,$$

where $X_{(a,b),w}$ denotes the number of co-occurrences of word $w$ with the syntactic word pair $(a, b)$ ($a$ denotes the noun/object) and $f(x) = \min(x, 100)$. This objective function isn't precisely targeting

---

[2]code for preprocessing, training, and experiments can be found at `https://github.com/abefrandsen/syntactic-rand-walk`

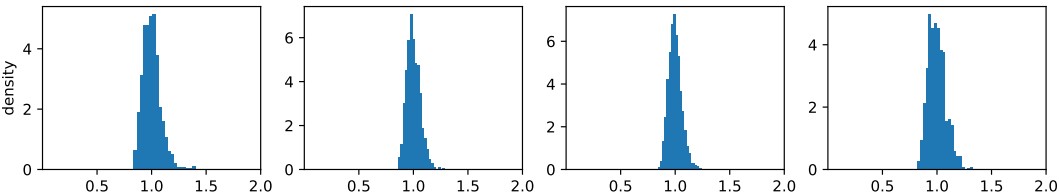

Figure 2: Histograms of partition functions $Z_{c,a}$ (x-axis is $Z_{c,a}/\mathbb{E}[Z_{c,a}]$)

the Tucker decomposition of the PMI3 tensor, but it is analogous to the training criterion used in Arora et al. (2015), and can be viewed as a negative log-likelihood for the model. To reduce the number of parameters, we constrain $T$ to have CP rank 1000. We also trained the embeddings and tensor jointly, but found that this approach yields very similar results. In all cases, we utilize the Tensorflow framework (Abadi et al., 2016) with the Adam optimizer (Kingma & Ba, 2014) (using default parameters), and train for 1-5 epochs.

## 5 EXPERIMENTAL VERIFICATION

In this section, we verify and evaluate our model empirically on select qualitative and quantitative tasks. In all of our experiments, we focus solely on syntactic word pairs formed by adjective-noun phrases, where the noun is considered the root word.

### 5.1 MODEL VERIFICATION

Arora et al. (2015) empirically verify the model assumptions of RAND-WALK, and since we trained our embeddings in the same way, we don't repeat their verifications here. Instead, we verify two key properties of syntactic RAND-WALK.

**Norm of composition tensor** We check the assumptions that the tensor $T$ is $(K, \epsilon)$-bounded. Ranging over all adjective-noun pairs in the corpus, we find that $\frac{1}{d}\|T(v_a, \cdot, \cdot) + I\|^2$ has mean 0.052 and maximum 0.248, $\frac{1}{d}\|T(v_a, \cdot, \cdot) + I\|_F^2$ has mean 1.61 and maximum 3.23, and $\frac{1}{d}\|T(v_a, v_b, \cdot)\|^2$ has mean 0.016 and maximum 0.25. Each of these three quantities has a well-bounded mean, but $\|T(v_a, \cdot, \cdot) + I\|^2$ has some larger outliers. If we ignore the log factors (which are likely due to artifacts in the proof) in Definition 2, the tensor is $(K, \epsilon)$ bounded for $K = 4$ and $\epsilon = 0.25$.

**Concentration of partition functions** In addition to Definition 2, we also directly check its implications: our model predicts that the partition functions $Z_{c,a}$ concentrate around their means. To check this, given a noun $a$, we draw 1000 random vectors $c$ from the unit sphere, and plot the histogram of $Z_{c,a}$.Results for a few randomly selected words $a$ are given in Figure 2. All partition functions that we inspected exhibited good concentration.

### 5.2 QUALITATIVE ANALYSIS OF COMPOSITION

We test the performance of our new composition for adjective-noun and verb-object pairs by looking for the words with closest embedding to the composed vector. For a phrase $(a, b)$, we compute $c = v_a + v_b + T(v_a, v_b, \cdot)$, and then retrieve the words $w$ whose embeddings $v_w$ have the largest cosine similarity to $c$. We compare our results to the additive composition method. Tables 1 and 2 show results for three adjective-noun and verb-object phrases. In each case, the tensor composition is able to retrieve some words that are more specifically related to the phrase. However, the tensor composition also sometimes retrieves words that seem unrelated to either word in the phrase. We conjecture that this might be due to the sparseness of co-occurrence of three words. We also observed cases where the tensor composition method was about on par with or inferior to the additive composition method for retrieving relevant words, particularly in the case of low-frequency phrases. More results can be found in supplementary material.

Table 1: Top 10 words relating to various adjective-noun phrases

| civil war | | complex numbers | | national park | |
|---|---|---|---|---|---|
| additive | tensor | additive | tensor | additive | tensor |
| war | civil | complex | complex | national | yosemite |
| civil | somalian | numbers | eigenvalues | park | denali |
| military | eicher | number | numbers | parks | gunung |
| army | crimean | function | hermitian | recreation | kenai |
| conflict | laotian | complexes | quaternions | forest | nps |
| wars | francoist | functions | marginalia | historic | teton |
| fought | ulysses | integers | azadi | heritage | refuges |
| revolutionary | liberian | multiplication | rationals | wildlife | tilden |
| forces | confederate | algebraic | holomorphic | memorial | snowdonia |
| outbreak | midst | integer | rhythmically | south | jigme |

Table 2: Top 10 words relating to various verb-object phrases

| took place | | took part | | took lead | |
|---|---|---|---|---|---|
| additive | tensor | additive | tensor | additive | tensor |
| place | occurred | part | participated | took | equalised |
| took | scheduled | took | participating | lead | halftime |
| death | commenced | taking | participate | taking | nailing |
| take | event | take | culminated | take | kenseth |
| taking | events | taken | organised | went | fumbled |
| birth | culminated | takes | participation | led | touchdown |
| taken | thursday | became | hostilities | taken | furlongs |
| takes | friday | came | culminating | came | trailed |
| came | postponed | put | invasion | put | keselowski |
| held | lasted | whole | undertook | wanted | peloton |

### 5.3 PHRASE SIMILARITY

We also test our tensor composition method on a adjective-noun phrase similarity task using the dataset introduced by Mitchell & Lapata (2010). The data consists of 108 pairs each of adjective-noun and verb-object phrases that have been given similarity ratings by a group of 54 humans. The task is to use the word embeddings to produce similarity scores that correlate well with the human scores; we use both the Spearman rank correlation and the Pearson correlation as evaluation metrics for this task. We note that the human similarity judgments are somewhat noisy; intersubject agreement for the task is $0.52$ as reported in Mitchell & Lapata (2010).

Given a phrase $(a, b)$ with embeddings $v_a, v_b$, respectively, we found that the tensor composition $v_a + v_b + T(v_a, v_b, \cdot)$ yields worse performance than the simple additive composition $v_a + v_b$. For this reason, we consider a *weighted* tensor composition $v_a + v_b + \alpha T(v_a, v_b, \cdot)$ with $\alpha \geq 0$. Following Mitchell & Lapata (2010), we split the data into a development set of 18 humans and a test set of the remaining 36 humans. We use the development set to select the optimal scalar weight for the weighted tensor composition, and using this fixed parameter, we report the results using the test set. We repeat this three times, rotating over folds of 18 subjects, and report the average results.

As a baseline, we also report the average results using just the additive composition, as well as a weighted additive composition $\beta v_a + v_b$, where $\beta \geq 0$. We select $\beta$ using the development set ("weighted1") and the test set ("weighted2"). We allow weighted2 to cheat in this way because it provides an upper bound on the best possible weighted additive composition. Additionally, we compare our method to the smoothed inverse frequency ("sif") weighting method that has been demonstrated to be near state-of-the-art for sentence embedding tasks (Arora et al., 2016). We also test embeddings of the form $p + \gamma \omega_a \omega_b T(v_a, v_b, \cdot)$ ("sif+tensor"), where $p$ is the sif embedding for $(a, b)$, $\omega_a$ and $\omega_b$ are the smoothed inverse frequency weights used in the sif embeddings, and $\gamma$ is a positive weight selected using the development set. The motivation for this hybrid embedding is to evaluate the extent to which the sif embedding and tensor component can independently improve performance on this task.

We perform these same experiments using two other standard sets of pre-computed word embeddings, namely GloVe[3] and carefully optimized cbow vectors[4] (Mikolov et al., 2017). We re-trained the composition tensor using the same corpus and technique as before, but substituting these pre-computed embeddings in place of the RAND-WALK (rw) embeddings. However, a bit of care must be taken here, since our syntactic RAND-WALK model constrains the norm of the word embeddings to be related to the frequency of the words, whereas this is not the case with the pre-computed embeddings. To deal with this, we rescaled the pre-computed embeddings sets to have the same norms as their counterparts in the rw embeddings, and then trained the composition tensor using these rescaled embeddings. At test time, we use the *original* embeddings to compute the additive components of our compositions, but use the *rescaled* versions when computing the tensor components.

The results for adjective-noun phrases are given in Tables 3. We observe that the tensor composition outperforms the additive compositions on all embedding sets apart from the Spearman correlation on the cbow vectors, where the weighted additive 2 method has a slight edge. The sif embeddings outperform the additive and tensor methods, but combining the sif embeddings and the tensor components yields the best performance across the board, suggesting that the composition tensor captures additional information beyond the individual word embeddings that is useful for this task. There was high consistency across the folds for the optimal weight parameter $\alpha$, with $\alpha = 0.4$ for the rw embeddings, $\alpha = .2, .3$ for the glove embeddings, and $\alpha = .3$ for the cbow embeddings. For the sif+tensor embeddings, $\gamma$ was typically in the range $[.1, .2]$.

The results for verb-object phrases are given in Table 4. Predicting phrase similarity appears to be harder in this case. Notably, the sif embeddings perform worse than unweighted vector addition. As before, we can improve the sif embeddings by adding in the tensor component. The tensor composition method achieves the best results for the glove and cbow vectors, but weighted addition works best for the randwalk vectors.

Overall, these results demonstrate that the composition tensor can improve the quality of the phrase embeddings in many cases, and the improvements are at least somewhat orthogonal to improvements

---

[3]obtained from `https://nlp.stanford.edu/projects/glove/`
[4]obtained from `https://fasttext.cc/docs/en/english-vectors.html`

Table 3: Correlation measures between human judgments and embedding-based similarity scores (Spearman, Pearson) for adjective-noun phrases across three embedding sets (top scores in each row are bolded)

|       | additive   | weighted1  | weighted2  | tensor     | sif        | sif+tensor       |
|-------|------------|------------|------------|------------|------------|------------------|
| rw    | .446, .438 | .444, .448 | .452, .453 | .460, .465 | .482, .477 | **.482**, **.481** |
| glove | .357, .336 | .351, .334 | .358, .345 | .368, .347 | .429, .434 | **.433**, **.437** |
| cbow  | .471, .452 | .469, .451 | .476, .456 | .474, .471 | .489, .482 | **.492**, **.484** |

Table 4: Correlation measures between human judgments and embedding-based similarity scores (Spearman, Pearson) for verb-object phrases

|       | additive   | weighted1  | weighted2          | tensor           | sif        | sif+tensor       |
|-------|------------|------------|--------------------|------------------|------------|------------------|
| rw    | .379, .370 | .391, .385 | **.392**, **.387** | .379, .370       | .378, .351 | .378, .363       |
| glove | .397, .400 | .398, .404 | .401, .404         | .410, **.420**   | .387, .380 | **.411**, .409   |
| cbow  | .423, .414 | .423, .410 | **.428**, .415     | **.428**, **.422** | .404, .404 | .420, .417       |

resulting from the sif embedding method. This suggests that a well-trained composition tensor used in conjunction with high quality word embeddings and additional embedding composition techniques has the potential to improve performance in downstream NLP tasks.

## ACKNOWLEDGMENTS

We thank Yingyu Liang, Mohit Bansal, and Eric Bailey for helpful discussions. Support from NSF CCF-1704656 is gratefully acknowledged.

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

Table 5: Top 10 words relating to various verb-object phrases

| giving birth | | solve problem | | changing name | |
| --- | --- | --- | --- | --- | --- |
| additive | tensor | additive | tensor | additive | tensor |
| birth | stillborn | problem | analytically | name | rebrand |
| giving | unborn | solve | creatively | changing | refocus |
| place | pregnant | problems | solve | change | redevelop |
| death | fathered | solving | subconsciously | changed | rebranding |
| give | litters | solved | devising | names | forgo |
| date | childbirth | solves | devise | referring | divest |
| gave | remarry | understand | proactively | title | rechristened |
| summary | newborn | resolve | solvers | word | afresh |
| gives | gestation | solution | extrapolate | actually | rebranded |
| given | eloped | question | rationalize | something | opting |

Table 6: Top 10 words relating to various adjective-noun phrases

| united states | | soviet union | | european union | |
| --- | --- | --- | --- | --- | --- |
| additive | tensor | additive | tensor | additive | tensor |
| united | united | union | union | european | eec |
| states | states | soviet | soviet | union | ebu |
| us | emigrating | ussr | sfsr | europe | dismemberment |
| canada | emirates | russian | disintegration | countries | retort |
| countries | immigrated | communist | lyudmila | federation | detracts |
| california | cartographic | russia | dismemberment | nations | arguable |
| usa | extradited | soviets | brezhnev | soviet | kely |
| america | senate | moscow | ussr | organisations | eea |
| kingdom | lighthouses | sfsr | perestroika | socialist | geosciences |
| nations | stateside | ukraine | zhukov | eu | bugzilla |

## A  ADDITIONAL QUALITIATIVE RESULTS

In this section we present additional qualitiative results demonstrating the use of the composition tensor for the retrieval of words related to adjective-noun and verb-object phrases.

In Table 5, we show results for the phrases "giving birth", "solve problem", and "changing name". These phrases are all among the top 500 most frequent verb-object phrases appearing in the training corpus. In these examples, the tensor-based phrase embeddings retrieve words that are generally markedly more related to the phrase at hand, and there are no strange false positives. These examples demonstrate how a verb-object phrase can encompass an action that isn't implied simply by the object or verb alone. The additive composition doesn't capture this action as well as the tensor composition.

Moving on to adjective-noun phrases, in Table 6, we show results for the phrases "United States", "Soviet Union", and "European Union". These phrases, which all occur with comparatively high frequency in the corpus, were identified as adjective-noun phrases by the tagger, but they function more as compound proper nouns. In each case, the additive composition retrieves reasonably relevant words, while the tensor composition is more of a mixed bag. In the case of "European Union", the tensor composition does retrieve the highly relevant words eec (European Economic Community) and eea (European Economic Area), which the additive composition misses, but the tensor composition also produces several false positives. It seems that for these types of phrases, the additive composition is sufficient to capture the meaning.

In Table 7, we fix the noun "taste" and vary the modifying adjective to highlight different senses of the noun. In the case of "expensive taste", both compositions retrieve words that seem to be either related to "expensive" or "taste", but there don't seem to be words that are intrinsically related to the phrase as a whole (with the exception, perhaps, of "luxurious", which the tensor composition

Table 7: Top 10 words relating to various adjective-noun phrases

| expensive taste | | awful taste | | refined taste | |
| additive | tensor | additive | tensor | additive | tensor |
| --- | --- | --- | --- | --- | --- |
| taste | expensive | taste | taste | taste | refined |
| expensive | taste | awful | awful | refined | taste |
| cheaper | costly | smell | smell | flavor | sweeter |
| flavor | prohibitively | unpleasant | disagreeable | tastes | sensuous |
| tastes | computationally | flavor | fruity | smell | elegant |
| unpleasant | cheaper | refreshing | aroma | flavour | disagreeable |
| inexpensive | luxurious | something | fishy | aroma | elegance |
| smell | sweeter | things | pungent | sour | neoclassicism |
| costly | inexpensive | really | odor | ingredients | refinement |
| ingredients | afford | odor | becuase | qualities | perfected |

Table 8: Top 10 words relating to various adjective-noun phrases

| close friend | | best friend | | dear friend | |
| additive | tensor | additive | tensor | additive | tensor |
| --- | --- | --- | --- | --- | --- |
| close | confidante | best | confidante | friend | friend |
| friend | confidant | friend | confides | dear | confidante |
| friends | coworker | actor | misinterpreting | colleague | coworker |
| confidant | close | awards | coworker | lover | colleague |
| colleague | friend | actress | memoirists | friends | dear |
| closest | confided | award | protege | girlfriend | confidant |
| collaborator | schoolmates | nominated | presumes | beloved | dearest |
| confidante | classmate | friends | helpfully | boyfriend | protege |
| classmate | protege | girlfriend | matth | classmate | confided |
| brother | cuz | writer | regretfully | roommate | collaborator |

retrieves). In the case of "awful taste", both compositions retrieve fairly similar words, which mostly relate to the physical sense of taste (rather than the more abstract sense of the word). For the phrase "refined taste", the additive composition fails to capture the sense of the phrase and retrieves many words related to food taste (which are irrelevant in this context), whereas the tensor composition retrieves more relevant words.

In Table 8, we fix the noun "friend" and vary the modifying adjective, but in all three cases, the adjective-noun phrase has basically the same meaning. In the case of "close friend" and "dear friend", both compositions retrieve fairly relevant and similar words. In the case of "best friend", both compositions retrieve false positives: the additive composition seems to find words related to movie awards, while the tensor composition finds unintuitive false positives. We note that in all three phrases, the tensor composition consistently retrieves the words "confidante", "confided" or "confides", "coworker", and "protoge", all of which are fairly relevant.

## A.1 Sentiment analysis

We test the effect of using the composition tensor for a sentiment analysis task. We use the movie review dataset of Pang and Lee (Pang & Lee, 2004) as well as the Large Movie Review dataset (Maas et al., 2011), which consist of 2,000 movie reviews and 50,000 movie reviews, respectively. For a fixed review, we identify each adjective-noun pair $(a, b)$ and compute $T(v_a, v_b, \cdot)$. We add these compositions together with the word embeddings for all of the words in the review, and then normalize the resulting sum. This vector is used as the input to a regularized logistic regression classifier, which we train using scikit-learn (Pedregosa et al., 2011) with the default parameters. We also consider a baseline method where we simply add together all of the word embeddings in the movie review, and then normalize the sum. We evaluate the test accuracy of each method using

Table 9: Test accuracy for sentiment analysis task (standard deviation reported in parentheses)

| Dataset | Additive | Tensor |
|---|---|---|
| Pang and Lee | 0.741 (0.018) | 0.759 (0.025) |
| Large Movie Review | 0.793 | 0.794 |

5-fold cross-validation on the smaller dataset and the training-test set split provided in the larger dataset. Results are shown in Table 9. Although the tensor method seems to have a slight edge over the baseline, the differences are not significant.

## B    OMITTED PROOFS FOR SECTION 3

In this section we will prove the main Theorem 1, which establishes the connection between the model parameters and the correlations of pairs/triples of words. As we explained in Section 3, a crucial step is to analyze the partition function of the model and show that the partition functions are concentrated. We will do that in Section B.1. We then prove the main theorem in Section B.2. More details and some technical lemmas are deferred to Section B.3

### B.1    CONCENTRATION OF PARTITION FUNCTION

In this section we will prove concentrations of partition functions (Lemma 1). Recall that we need the tensor to be $K$-bounded (where $K$ is a constant) for this to work.

**Definition 3.** *(Definition 2 restated) The composition tensor $T$ is $(K, \epsilon)$-bounded, if for any word embedding $v_a, v_b$, we have*

$$\|T(v_a, \cdot, \cdot) + I\|^2 \leq \frac{Kd\epsilon^2}{\log^2 n}; \|T(v_a, \cdot, \cdot) + I\|_F^2 \leq Kd; \|T(v_a, v_b, \cdot)\|^2 \leq Kd.$$

Note that $K$ here should be considered as an absolute constant (like 5, in fact in Section 5 we show $K$ is less than 4). We first restate Lemma 1 here:

**Lemma 3** (Lemma 1 restated). *For the syntactic RAND-WALK model, there exists a constant $Z$ such that*

$$\Pr_{c \sim \mathcal{C}}[(1 - \epsilon_z)Z \leq Z_c \leq (1 + \epsilon_z)Z] \geq 1 - \delta,$$

*for $\epsilon_z = \tilde{O}(1/\sqrt{n})$ and $\delta = \exp(-\Omega(\log^2 n))$.*

*Furthermore, if the tensor $T$ is $(K, \epsilon)$-bounded, then for any fixed word $a \in V$, there exists a constant $Z_a$ such that*

$$\Pr_{c \sim \mathcal{C}}[(1 - \epsilon_{z,a})Z_a \leq Z_{c,a} \leq (1 + \epsilon_{z,a})Z_a] \geq 1 - \delta,$$

*for $\epsilon_{z,a} = O(\epsilon) + \tilde{O}(1/\sqrt{n})$ and $\delta = \exp(-\Omega(\log^2 n))$.*

In fact, the first part of this Lemma is exactly Lemma 2.1 in Arora et al. (2015). Therefore we will focus on the proof of the second part.

For the second part, we know the probability of choosing a word $b$ is proportional to $\exp(T(v_a, v_b, c) + \langle c, v_b \rangle) = \exp(\langle T(v_a, \cdot, c) + c, v_b \rangle)$.

If the probability of choosing word $w$ is proportional to $\exp(\langle r, v_w \rangle)$ for some vector $r$ (think of $r = T(v_a, \cdot, c) + c$), then in expectation the partition function should be equal to $n\mathbb{E}_{v \sim \mathcal{D}_V}[\exp(\langle r, v \rangle)]$ (here $\mathcal{D}_V$ is the distribution of word embedding). When the number of words is large enough, we hope that with high probability the partition function is close to its expectation. Since the Gaussian distribution is spherical, we also know that the expected partition function $n\mathbb{E}_{v \sim \mathcal{D}_V}[\exp(\langle r, v \rangle)]$ should only depend on the norm of $r$. Therefore as long as we can prove the norm of $r = T(v_a, \cdot, c) + c$ remain similar for most $c$, we will be able to prove the desired result in the lemma.

We will first show the norm of $r = T(v_a, \cdot, c) + c$ is concentrated if the tensor $T$ is $(K, \epsilon)$-bounded. Throughout all subsequent proofs, we assume that $\epsilon < 1$ and $d \geq \log^2 n/\epsilon^2$.

**Lemma 4.** *Let $v_a$ be a fixed word vector, and let $c$ be a random discourse vector. If $T$ is $(K, \epsilon)$-bounded with $d \geq \log^2 n/\epsilon^2$, we have*

$$\Pr[\|T(v_a, \cdot, c) + c\|^2 \in L \pm O(\epsilon)] \geq 1 - \delta,$$

*where $0 \leq L \leq K$ is a constant that depends on $v_a$, and $\delta = \exp(-\Omega(\log^2 n))$.*

*Proof.* Since $c$ is a uniform random vector on the unit sphere, we can represent $c$ as $c = z/\|z\|$, where $z \sim N(0, I)$ is a standard spherical Gaussian vector. For ease of notation, let $M = T(v_a, \cdot, \cdot) + I$, and write the singular value decomposition of $M$ as $M = U\Sigma V^T$. Note that $\Sigma = \text{diag}(\lambda_1, \ldots, \lambda_d)$ and $U$ and $V$ are orthogonal matrices, so that in particular, the random variable $y = V^T z$ has the same distribution as $z$, i.e. its entries are i.i.d. standard normal random variables. Further, $\|Ux\|^2 = \|x\|^2$ for any vector $x$, since $U$ is orthogonal. Hence, we have

$$\|T(v_a, \cdot, c) + c\|^2 = \frac{1}{\|z\|^2}\|Mz\|^2 = \frac{1}{\|z\|^2}\|U\Sigma y\|^2 = \frac{\sum_{i=1}^d \lambda_i^2 y_i^2}{\sum_{i=1}^d z_i^2}.$$

Since both the numerator and denominator of this quantity are generalized $\chi^2$ random variables, we can apply Lemma 7 to get tail bounds on both. Observe that by assumption, we have $\lambda_i^2 \leq K d\epsilon^2/\log^2 n$ for all $i$, and $\sum_{i=1}^d \lambda_i^2 \leq Kd$. Set $A = \sum_{i=1}^d \lambda_i^2 y_i^2$ and $B = \sum_{i=1}^d z_i^2$. Let $\lambda_{max}^2 = \max_{1 \leq i \leq d} \lambda_i^2$. Note that $\mathbb{E}[A] = \sum_{i=1}^d \lambda_i^2 \leq Kd$ and $\mathbb{E}[B] = d$.

We will apply Lemma 7 to prove concentration bounds for $A$, in this case we have

$$\Pr\left[|A - \mathbb{E}[A]| \geq 2\sqrt{\sum_{i=1}^d \lambda_i^4}\sqrt{x} + 2\lambda_{max}^2 x\right] \leq 2\exp(-x).$$

Under our assumptions, we know $\lambda_{max}^2 \leq Kd\epsilon^2/\log^2 n$ and $\sqrt{\sum_{i=1}^d \lambda_i^4} \leq \sqrt{\lambda_{max}^2 \sum_{i=1}^d \lambda_i^2} \leq Kd\epsilon/\log n$. Take $x = \frac{1}{16}\log^2 n$, we know $2\sqrt{\sum_{i=1}^d \lambda_i^4}\sqrt{x} + 2\lambda_{max}^2 x] \leq Kd\epsilon$. Therefore

$$\Pr[|A - \mathbb{E}[A]| \geq Kd\epsilon] \leq 2\exp(-\Omega(\log^2 n)).$$

Similarly, we can apply Lemma 7 to $B$ (in fact we can apply simpler concentration bounds for standard $\chi^2$ distribution), and we get

$$\Pr[|B - \mathbb{E}[B]| \geq 2\sqrt{d}\sqrt{x} + 2x] \leq 2\exp(-x).$$

If we take $x = \frac{1}{16}\log^2 n$, we know $2\sqrt{d}\sqrt{x} + 2x \leq \epsilon d$. This implies

$$\Pr[|B - \mathbb{E}[B]| \geq d\epsilon] \leq 2\exp(-\Omega(\log^2 n)).$$

When both events happen we know $|\frac{A}{B} - \frac{\mathbb{E}[A]}{\mathbb{E}[B]}| \leq 4K\epsilon = O(\epsilon)$ (here $K$ is considered as a constant). This finishes the proof.

$\square$

Using this lemma, we will show that the expected condition number $n\mathbb{E}_{v \sim \mathcal{D}_V}[\exp(\langle r, v \rangle)]$ (where $r = T(v_a, \cdot, c) + c$) is concentrated

**Lemma 5.** *Let $v_a$ be a fixed word vector, and let $c$ be a random discourse vector. If $T$ is $(K, \epsilon)$-bounded, there exists $Z_a$ such that we have*

$$\Pr[n\mathbb{E}_{v \sim \mathcal{D}_V}[\exp(\langle T(v_a, \cdot, c) + c, v \rangle)] \in Z_a(1 \pm O(\epsilon) \geq 1 - \delta,$$

*where $Z_a = \Theta(n)$ depends on $v_a$, and $\delta = \exp(-\Omega(\log^2 n))$.*

*Proof.* We know $v = s \cdot \hat{v}$ where $\hat{v} \sim N(0, I)$ and $s$ is a (random) scaling. Let $r = T(v_a, \cdot, c) + c$. Conditioned on $s$ we know $\langle r, v \rangle$ is equivalent to a Gaussian random variable with standard deviation $\sigma = \|r\|s$. For this random variable we know

$$
\begin{aligned}
\mathbb{E}[\exp(\langle r, v \rangle)|s] &= \int_x \frac{1}{\sigma\sqrt{2\pi}} \exp\left(-\frac{x^2}{2\sigma^2}\right) \exp(x) dx \\
&= \int_x \frac{1}{\sigma\sqrt{2\pi}} \exp\left(-\frac{(x - \sigma^2)^2}{2\sigma^2} + \sigma^2/2\right) dx \\
&= \exp(\sigma^2/2).
\end{aligned}
$$

Hence,
$$
\mathbb{E}[\exp(\langle r, v \rangle)|s] = \exp(s^2 \|r\|^2/2).
$$

Let $g(x) = \mathbb{E}_s[\exp(s^2 x/2)]$, we know $g'(x) = \mathbb{E}_s[\exp(s^2 x/2) \cdot (s^2/2)] \leq \kappa^2/2 \cdot g(x)$. In particular, this implies $g(x + \gamma) \leq \exp(\kappa^2 \gamma/2) g(x)$ (for small $\gamma$).

By Lemma 4, we know with probability at least $1 - \Omega(\log^2 n)$, $\|r\|^2 \in L \pm O(\epsilon)$. Therefore, when this holds, we have
$$
n\mathbb{E}_{v \sim \mathcal{D}_V}[\exp(\langle r, v \rangle)] \in ng(L - O(\epsilon)) \cdot [1, \exp(O(\epsilon\kappa^2/2)].
$$

The multiplicative factor on the RHS is bounded by $1 + O(\epsilon)$ when $\epsilon$ is small enough (and $\kappa$ is a constant). This finishes the proof. $\square$

Now we know the expected partition function is concentrated (for almost all discourse vectors $c$), it remains to show when we have finitely many words the partition function is concentrated around its expectation. This was already proved in Arora et al. (2015), we use their lemma below:

**Lemma 6.** *For any fixed vector $r$ (whose norm is bounded by a constant), with probability at least $1 - \exp(-\Omega(\log^2 n))$ over the choices of the words, we have*

$$
\sum_{i=1}^n \exp(\langle r, v_i \rangle) \in n\mathbb{E}_{v \sim \mathcal{D}_V}[\exp(\langle r, v \rangle)](1 \pm \epsilon_z),
$$

*where $\epsilon_z = \tilde{O}(1/\sqrt{n})$.*

This is essentially Lemma 2.1 in Arora et al. (2015) (see Equation A.32). The version we stated is a bit different because we allow $r$ to have an arbitrary constant norm (while in their proof vector $r$ is the discourse vector $c$ and has norm 1). This is a trivial corollary as we can move the norm of $r$ into the distribution of the scaling factor $s$ for the word embedding.

Finally we are ready to prove Lemma 1.

*Proof of Lemma 1.* The first part is exactly Lemma 2.1 in Arora et al. (2015).

For the second part, note that the partition function $Z_{c,a} = \sum_{i=1}^n \langle T(v_a, \cdot, c) + c, v_i \rangle$. We will use $\mathbb{E}[Z_{c,a}]$ to denote its expectation over the randomness of the word embedding $\{v_i\}$. By Lemma 5, we know for at least $1 - \exp(-\Omega(\log^2 n))$ fraction of discourse vectors $c$, the expected partition function is concentrated ($\mathbb{E}[Z_{c,a}] \in (1 \pm O(\epsilon))Z_a$). Let $\mathcal{S}$ denote the set of $c$ such that Lemma 5 holds. Now by Lemma 6 we know for any $x \in \mathcal{S}$, with probability at least $1 - \exp(-\Omega(\log^2 n))$ $Z_{c,a} \in (1 \pm \epsilon_z)\mathbb{E}[Z_{c,a}]$.

Therefore we know if we consider both $c$ and the embedding as random variables, $\Pr[Z_{c,a} \in (1 \pm O(\epsilon + \epsilon_z))Z_a] \geq 1 - \delta'$ where $\delta' = \exp(-\Omega(\log^2 n))$. Let $S$ be the set of word embedding such that there is at least $\sqrt{\delta'}$ fraction of $c$ that does not satisfy $Z_{c,a} \in (1 \pm O(\epsilon + \epsilon_z))Z_a$, we must have $\Pr[S] \cdot \sqrt{\delta'} \leq \delta'$. Therefore
$$
\Pr[S] \leq \sqrt{\delta'}.
$$

That is, with probability at least $1 - \sqrt{\delta'}$ (over the word embeddings), there is at least $1 - \sqrt{\delta'}$ fraction of $c$ such that $Z_{c,a} \in (1 \pm O(\epsilon + \epsilon_z))Z_a$.

$\square$

## B.2 ESTIMATING THE CORRELATIONS

In this section we prove Theorem 1 and Corollary 1. The proof is very similar to the proof of Theorem 2.2 in Arora et al. (2015). We use several lemmas in that proof, and these lemmas are deferred to Section B.3.

*Proof of Theorem 1.* Throughout this proof we consider two adjacent discourse vectors $c, c'$, where $c$ generated a single word $w$ and $c'$ generated a syntactic pair $(a, b)$.

The first two results in Theorem 1 are exactly the same as Theorem 2.2 in Arora et al. (2015). Therefore we only need to prove the result for $p([a, b])$ and $p(w, [a, b])$.

For $p([a, b])$, by definition of the model we know

$$p([a, b]) = \mathbb{E}_{c'}[\frac{1}{Z_{c'}} \frac{1}{Z_{c',a}} \exp(\langle c', v_a \rangle + \langle c', v_b \rangle + T(v_a, v_b, c'))].$$

Here $Z_{c'}$ is the partition function $\sum_{i=1}^{n} \exp(\langle c', v_i \rangle)$, and $Z_{c',a}$ is the partition function $\sum_{i=1}^{n} \exp(\langle c', v_i \rangle + T(v_a, v_i, c'))$.

Let $\mathcal{F}$ be the event that $c'$ satisfies the equations in Lemma 1. Let $\bar{\mathcal{F}}$ be its negation. By Lemma 1 we know $\Pr[\mathcal{F}] \geq 1 - \exp(-\Omega(\log^2 n))$. Using this event, we can write

$$p([a, b]) = \mathbb{E}_{c'}[\frac{1}{Z_{c'}} \frac{1}{Z_{c',a}} \exp(\langle c', v_a \rangle + \langle c', v_b \rangle + T(v_a, v_b, c'))1_{\mathcal{F}}]$$
$$+ \mathbb{E}_{c'}[\frac{1}{Z_{c'}} \frac{1}{Z_{c',a}} \exp(\langle c', v_a \rangle + \langle c', v_b \rangle + T(v_a, v_b, c'))1_{\bar{\mathcal{F}}}].$$

The second term can be bounded by Lemma 8 and the fact that $Z_{c'} Z_{c',a} \geq \beta$ from Lemma 2. We know

$$\mathbb{E}_{c'}[\frac{1}{Z_{c'}} \frac{1}{Z_{c',a}} \exp(\langle c', v_a \rangle + \langle c', v_b \rangle + T(v_a, v_b, c')1_{\bar{\mathcal{F}}}] \leq \exp(-\Omega(\log^{1.8} n)).$$

For the first term, we know by Lemma 1 that there exists $Z, Z_a$ that are close to $Z_{c'}$ and $Z_{c',a}$. Therefore

$$p([a, b]) = \mathbb{E}_{c'}[\frac{1}{Z_{c'}} \frac{1}{Z_{c',a}} \exp(\langle c', v_a \rangle + \langle c', v_b \rangle + T(v_a, v_b, c')1_{\mathcal{F}}]$$
$$+ \mathbb{E}_{c'}[\frac{1}{Z_{c'}} \frac{1}{Z_{c',a}} \exp(\langle c', v_a \rangle + \langle c', v_b \rangle + T(v_a, v_b, c')1_{\bar{\mathcal{F}}}].$$
$$\leq (1 + \epsilon_z)(1 + \epsilon_{z,a}) \mathbb{E}_{c'}[\frac{1}{Z} \frac{1}{Z_a} \exp(\langle c', v_a \rangle + \langle c', v_b \rangle + T(v_a, v_b, c')1_{\mathcal{F}}] + \exp(-\Omega(\log^{1.8} n))$$
$$\leq \frac{(1 + \epsilon_z)(1 + \epsilon_{z,a})}{Z Z_a} \mathbb{E}_{c'}[\exp(\langle c', v_a \rangle + \langle c', v_b \rangle + T(v_a, v_b, c'))] + \exp(-\Omega(\log^{1.8} n))$$
$$\leq \frac{(1 + \epsilon_z)(1 + \epsilon_{z,a})(1 + \tilde{O}(1/d))}{Z Z_a} \exp(\frac{\|v_a + v_b + T(v_a, v_b, \cdot)\|^2}{2d}) + \exp(-\Omega(\log^{1.8} n)).$$

Here the last step used Lemma 10. Since both $Z$ and $Z_a$ can be bounded by $O(n)$, and $\frac{\|v_a + v_b + T(v_a, v_b, \cdot)\|^2}{2d}$ is bounded by $(4\kappa + \sqrt{2K})^2$, we know the first term is of order $\Omega(1/n^2)$, and the second term is negligible.

For the lowerbound, we can have

$$p([a,b]) = \mathbb{E}_{c'}[\frac{1}{Z_{c'}}\frac{1}{Z_{c',a}}\exp(\langle c', v_a\rangle + \langle c', v_b\rangle + T(v_a, v_b, c')1_{\mathcal{F}}]$$

$$+ \mathbb{E}_{c'}[\frac{1}{Z_{c'}}\frac{1}{Z_{c',a}}\exp(\langle c', v_a\rangle + \langle c', v_b\rangle + T(v_a, v_b, c')1_{\bar{\mathcal{F}}}].$$

$$\geq (1 - \epsilon_z)(1 - \epsilon_{z,a})\mathbb{E}_{c'}[\frac{1}{Z}\frac{1}{Z_a}\exp(\langle c', v_a\rangle + \langle c', v_b\rangle + T(v_a, v_b, c')1_{\mathcal{F}}]$$

$$\geq \frac{(1 - \epsilon_z)(1 - \epsilon_{z,a})}{ZZ_a}\{\mathbb{E}_{c'}[\exp(\langle c', v_a\rangle + \langle c', v_b\rangle + T(v_a, v_b, c'))]$$

$$-\mathbb{E}_{c'}[\exp(\langle c', v_a\rangle + \langle c', v_b\rangle + T(v_a, v_b, c'))1_{\bar{\mathcal{F}}}]\}$$

$$\geq \frac{(1 - \epsilon_z)(1 - \epsilon_{z,a})}{ZZ_a}\{\mathbb{E}_{c'}[\exp(\langle c', v_a\rangle + \langle c', v_b\rangle + T(v_a, v_b, c'))] - \exp(-\Omega(\log^{1.8} n))\}$$

$$\geq \frac{(1 - \epsilon_z)(1 - \epsilon_{z,a})(1 - \tilde{O}(1/d))}{ZZ_a}\left\{\exp(\frac{\|v_a + v_b + T(v_a, v_b, \cdot)\|^2}{2d}) - \exp(-\Omega(\log^{1.8} n))\right\}.$$

Again the last step is using Lemma 10 and the term $\exp(-\Omega(\log^{1.8} n))$ is negligible. Combining the upper and lower bound, we know

$$\log p([a,b]) = \frac{\|v_a + v_b + T(v_a, v_b, \cdot)\|^2}{2d} - \log Z - \log Z_a \pm \epsilon_p,$$

where $\epsilon_p = O(\epsilon_z + \epsilon_{z,a}) + \tilde{O}(1/d)$.

Now we turn to the most complicated term $\log p(w, [a,b])$. By definition we know

$$p(w, [a,b]) = \mathbb{E}_{c,c'}[\frac{1}{Z_c}\exp(\langle c, v_w\rangle)\frac{1}{Z_{c'}}\frac{1}{Z_{c',a}}\exp(\langle c', v_a\rangle + \langle c', v_b\rangle + T(v_a, v_b, c'))].$$

We will follow similar idea as before. Let $\mathcal{F}$ be the event that both $c, c'$ satisfy the equations in Lemma 1 and $\bar{\mathcal{F}}$ be its negation. By Lemma 1 and union bound we know $\Pr[\mathcal{F}] \geq 1 - \exp(-\Omega(\log^2 n))$.

We again separate the co-occurrence probability based on the event $\mathcal{F}$:

$$p(w, [a,b]) = \mathbb{E}_{c,c'}[\frac{1}{Z_c}\exp(\langle c, v_w\rangle)\frac{1}{Z_{c'}}\frac{1}{Z_{c',a}}\exp(\langle c', v_a\rangle + \langle c', v_b\rangle + T(v_a, v_b, c')1_{\mathcal{F}}]$$

$$+ \mathbb{E}_{c,c'}[\frac{1}{Z_c}\exp(\langle c, v_w\rangle)\frac{1}{Z_{c'}}\frac{1}{Z_{c',a}}\exp(\langle c', v_a\rangle + \langle c', v_b\rangle + T(v_a, v_b, c')1_{\bar{\mathcal{F}}}].$$

For the second term, we can again use Lemma 8 to show that it is bounded by $\exp(-\Omega(\log^{1.8} n))$. Now, using techniques similar as before, we can prove

$$p(w, [a,b]) = (1 \pm O(\epsilon_z + \epsilon_{z,a}))\frac{1}{Z^2 Z_a}\mathbb{E}_{c,c'}[\exp(\langle c, v_w\rangle)\exp(\langle c', v_a\rangle + \langle c', v_b\rangle + T(v_a, v_b, c'))].$$

$$(9)$$

Now the final step is to use the fact that $c$ and $c'$ are close to simplify the final formula. Let $A(c') = \mathbb{E}_{c|c'}[\exp(\langle c, v_w\rangle)]$, by Lemma 9 we know $A(c') \in (1 \pm \epsilon_w)\exp(\langle v_w, c'\rangle)$. Therefore

$$\mathbb{E}_{c,c'}[\exp(\langle c, v_w\rangle)\exp(\langle c', v_a\rangle + \langle c', v_b\rangle + T(v_a, v_b, c'))]$$

$$=\mathbb{E}_{c'}[\exp(\langle c', v_a\rangle + \langle c', v_b\rangle + T(v_a, v_b, c'))\mathbb{E}_{c|c'}[\exp(\langle c, v_w\rangle)]]$$

$$=\mathbb{E}_{c'}[\exp(\langle c', v_a\rangle + \langle c', v_b\rangle + T(v_a, v_b, c'))A(c')]$$

$$=(1 \pm \epsilon_w)\mathbb{E}_{c'}[\exp(\langle c', v_a\rangle + \langle c', v_b\rangle + T(v_a, v_b, c') + \langle c', v_w\rangle)]$$

$$=(1 \pm \epsilon_w)(1 \pm \tilde{O}(1/d))\exp(\frac{\|v_w + v_a + v_b + T(v_a, v_b, \cdot)\|^2}{2d}).$$

Here the last step is again by Lemma 10. Combining this with Equation equation 9 gives the result. $\qquad\square$

Finally we prove Corollary 1, which is just a simple calculation based on Theorem 1:

*Proof of Corollary 1.* By the definition of PMI3, we know

$$PMI3 = \log p(w, [a, b]) + \log p(a) + \log p(b) + \log p(w) - \log p(w, a) - \log p(w, b) - \log p([a, b]).$$

$$= (\frac{\|v_w + v_a + v_b + T(v_a, v_b, \cdot)\|^2}{2d} - 2\log Z - \log Z_a) + (\frac{\|v_a\|^2}{2d} + \frac{\|v_b\|^2}{2d} + \frac{\|v_w\|^2}{2d} - 3\log Z)$$

$$- (\frac{\|v_w + v_a\|^2}{2d} + \frac{\|v_w + v_b\|^2}{2d} - 4\log Z) - (\frac{\|v_a + v_b + T(v_a, v_b, \cdot)\|^2}{2d} - \log Z - \log Z_a) \pm 7\epsilon$$

$$= \frac{T(v_a, v_b, v_w)}{d} \pm 7\epsilon.$$

$\square$

### B.3 AUXILIARY LEMMAS

**Tail bound for $\chi^2$ distribution** We will use the following tail bounds for the generalized $\chi^2$-squared distribution.

**Lemma 7.** *Laurent & Massart (2000) Let $y_1, \ldots, y_d$ be i.i.d. standard normal random variables, and let $a_1, \ldots, a_d$ be nonnegative real numbers. Set $Y = \sum_{i=1}^{d} a_i y_i^2$ and $a = (a_1, a_2, \ldots, a_d)$. Then the following hold for any positive real number $x$:*

$$P(Y - \mathbb{E}[Y] \geq 2\|a\|_2\sqrt{x} + 2\|a\|_\infty x) \leq \exp(-x)$$
$$P(Y - \mathbb{E}[Y] \leq -2\|a\|_2\sqrt{x}) \leq \exp(-x).$$

**Additional Lemmas** We will use several tools developed in Arora et al. (2015). The first lemma allows us to bound the probabilities the discourse vector $c$ does not satisfy the results of Lemma 1.

**Lemma 8.** *Let $\mathcal{F}$ be any event that depends on the discourse vector $c$ with probability at least $1 - \exp(-\Omega(\log^2 n))$, and $\bar{\mathcal{F}}$ be its negation. Suppose $r$ is a vector of norm $O(\sqrt{d})$, then*

$$\mathbb{E}_c[\exp(\langle r, c \rangle)1_{\bar{\mathcal{F}}}] \leq \exp(-\Omega(\log^{1.8} n)).$$

*Further, if we consider two consecutive discourse vectors $c$, $c'$, redefine $\mathcal{F}$ to be an event that can depend on both discourse vectors, again with probability at least $1 - \exp(-\Omega(\log^2 n))$. If $r, r'$ are two vectors of norm $O(\sqrt{d})$ we have*

$$\mathbb{E}_{c,c'}[\exp(\langle r, c \rangle)\exp(\langle r', c' \rangle)1_{\bar{\mathcal{F}}}] \leq \exp(-\Omega(\log^{1.8} n)).$$

*Proof.* The proof of this lemma appears on page 20 in Arora et al. (2015), as a step in the proof of their Theorem 2.2. For completeness, we reproduce (and slightly adapt) their argument here.

Observe that

$$\mathbb{E}_c[\exp(\langle r, c \rangle)1_{\bar{\mathcal{F}}}] = \mathbb{E}_c[\exp(\langle r, c \rangle)1_{\langle r,c \rangle > 0}1_{\bar{\mathcal{F}}}] + \mathbb{E}_c[\exp(\langle r, c \rangle)1_{\langle r,c \rangle < 0}1_{\bar{\mathcal{F}}}].$$

The second term of B.3 is upper bounded by

$$\mathbb{E}_c[1_{\bar{\mathcal{F}}}] \leq \exp(-\Omega(\log^2 n)).$$

Note that the first term of B.3 can be bounded as follows:

$$\mathbb{E}_c[\exp(\langle r, c \rangle)1_{\langle r,c \rangle > 0}1_{\bar{\mathcal{F}}}] \leq \mathbb{E}_c[\exp(\langle \alpha r, c \rangle)1_{\langle r,c \rangle > 0}1_{\bar{\mathcal{F}}}] \leq \mathbb{E}_c[\exp(\langle \alpha r, c \rangle)1_{\bar{\mathcal{F}}}]$$

for $\alpha > 1$. Therefore, to obtain a bound on $\mathbb{E}_c[\exp(\langle r, c \rangle)1_{\langle r,c \rangle > 0}1_{\bar{\mathcal{F}}}]$ is suffices to bound

$$\mathbb{E}_c[\exp(\langle r, c \rangle)1_{\bar{\mathcal{F}}}]$$

when $\|r\| = \Omega(\sqrt{d})$.

Let $z$ denote the random variable $\langle r, c \rangle$, and let $r(z) = 1_{\bar{\mathcal{F}}}$. Using Lemma A.4 in Arora et al. (2015), we have

$$\mathbb{E}_c[\exp(z)r(z)] \leq \mathbb{E}_c[\exp(z)1_{[t,\infty]}(z)],$$

where $t$ satisfies that $\mathbb{E}_c[1_{[t,\infty]}(z)] = \Pr[z \geq t] = E_c[r(z)] \leq \exp(-\Omega(\log^2 n))$. Then by Lemma A.1 of Arora et al. (2015), we have that $t \geq \Omega(\log^{.9} n)$. Finally, applying Corollary A.3 of Arora et al. (2015), we have

$$\mathbb{E}_c[\exp(z)r(z)] \leq E_c[\exp(z)1_{[t,\infty]}(z)] = \exp(-\Omega(\log^{1.8} n)),$$

which completes the proof for the first part of this lemma.

The second part of this lemma can be proved in much the same fashion. By Cauchy-Schwarz,

$$\left(\mathbb{E}_{c,c'}[\exp(\langle r, c \rangle)\exp(\langle r', c' \rangle)1_{\bar{\mathcal{F}}}]\right)^2 \leq \left(\mathbb{E}_{c,c'}[\exp(\langle r, c \rangle)^2 1_{\bar{\mathcal{F}}}]\right)\left(\mathbb{E}_{c,c'}[\exp(\langle r', c' \rangle)^2 1_{\bar{\mathcal{F}}}]\right)$$
$$\leq \left(\mathbb{E}_c[\exp(\langle 2r, c \rangle)\mathbb{E}_{c'|c}[1_{\bar{\mathcal{F}}}]]\right)\left(\mathbb{E}_{c'}[\exp(\langle 2r', c' \rangle)\mathbb{E}_{c|c'}[1_{\bar{\mathcal{F}}}]]\right).$$

Now we bound $\mathbb{E}_c[\exp(\langle 2r, c \rangle)\mathbb{E}_{c'|c}[1_{\bar{\mathcal{F}}}]]$ using the same argument as above in the first part of this proof, replacing $1_{\bar{\mathcal{F}}}$ with $\mathbb{E}_{c'|c}[1_{\bar{\mathcal{F}}}]$, $r$ with $2r$, and $r(z) = 1_{\bar{\mathcal{F}}}$ with $r(z) = \mathbb{E}_{c'|z}[1_{\bar{\mathcal{F}}}]$. In particular, we have $\mathbb{E}_c[\exp(\langle 2r, c \rangle)\mathbb{E}_{c'|c}[1_{\bar{\mathcal{F}}}]] \leq \exp(-\Omega(\log^{1.8} n))$. Likewise, we have the same bound for $\mathbb{E}_{c'}[\exp(\langle 2r', c' \rangle)\mathbb{E}_{c|c'}[1_{\bar{\mathcal{F}}}]]$. Putting these two together, we conclude that

$$\mathbb{E}_{c,c'}[\exp(\langle r, c \rangle)\exp(\langle r', c' \rangle)1_{\bar{\mathcal{F}}}] \leq \left(\mathbb{E}_c[\exp(\langle 2r, c \rangle)\mathbb{E}_{c'|c}[1_{\bar{\mathcal{F}}}]]\right)^{1/2}\left(\mathbb{E}_{c'}[\exp(\langle 2r', c' \rangle)\mathbb{E}_{c|c'}[1_{\bar{\mathcal{F}}}]]\right)^{1/2}$$
$$\leq \exp(-\Omega(\log^{1.8} n)),$$

as desired.

$\square$

The next lemma allows us to handle the difference between two consecutive discourse vectors:

**Lemma 9.** *Let $c, c'$ be two discourse vectors that are adjacent, let $v_w$ be a word embedding satisfying $\|v_w\| \leq K'\sqrt{d}$, and let $A(c) := \mathbb{E}_{c'|c}[\exp(\langle v_w, c' \rangle)]$, then we have*

$$A(c) \in (1 \pm \epsilon_w)\exp(\langle v_w, c \rangle).$$

*Proof.* The proof of this lemma appears on page 21 in Arora et al. (2015), again as a step in the proof of their Theorem 2.2. For completeness, we reproduce the argument here.

Since $\|v_w\| \leq K'\sqrt{d}$ for some constant $K'$, we have that $\langle v_w, c - c' \rangle \leq \|v_w\|\|c - c'\| \leq K'\sqrt{d}\|c - c'\|$. Hence,

$$A(c) = \mathbb{E}_{c'|c}[\exp(\langle v_w, c' \rangle)]$$
$$= \exp(\langle v_w, c \rangle)\mathbb{E}_{c'|c}[\exp(\langle v_w, c' - c \rangle)]$$
$$\leq \exp(\langle v_w, c \rangle)\mathbb{E}_{c'|c}[K'\sqrt{d}\|c - c'\|]$$
$$\leq (1 + \epsilon_w)\exp(\langle v_w, c \rangle),$$

where the last inequality follows from our model assumptions.

To get the lower bound, observe that

$$\mathbb{E}_{c'|c}[\exp(K'\sqrt{d}\|c - c'\|)] + \mathbb{E}_{c'|c}[\exp(-K'\sqrt{d}\|c - c'\|)] \geq 2.$$

Therefore, the model assumptions imply that

$$\mathbb{E}_{c'|c}[\exp(-K'\sqrt{d}\|c - c'\|)] \geq 1 - \epsilon_w.$$

Hence,

$$A(c) = \exp(\langle v_w, c \rangle)\mathbb{E}_{c'|c}[\exp(\langle v_w, c' - c \rangle)]$$
$$\geq \exp(\langle v_w, c \rangle)\mathbb{E}_{c'|c}[\exp(K'\sqrt{d}\|c - c'\|)]$$
$$\geq (1 - \epsilon_w)\exp(\langle v_w, c \rangle).$$

$\square$

The next lemma we use gives bound on $\mathbb{E}[\exp(\langle v, c \rangle)]$ where $c$ is a uniform vector on the unit sphere.

**Lemma 10.** *[Lemma A.5 in Arora et al. (2015)] Let $v \in \mathbb{R}^d$ be a fixed vector with norm $\|v\| = O(\sqrt{d})$. For random variable $c$ with uniform distribution over the sphere, we have that*

$$\log \mathbb{E}[\exp(\langle v, c\rangle)] = \frac{\|v\|^2}{2d} \pm \epsilon_c,$$

*where $\epsilon_c = \tilde{O}(1/d)$.*

We end with the proof of Lemma 2.

*Proof of Lemma 2.* Just for this proof, we use the following notation. Let $I_{d\times d}$ be the $d$-dimensional identity matrix, and let $x_1, x_2, \ldots, x_n$ be i.i.d. draws from $N(0, I_{d\times d})$. Let $y_i = \|x_i\|_2$, and note that $y_i^2$ is a standard $\chi$-squared random variable with $d$ degrees of freedom. Let $\kappa$ be a positive constant, and let $s_1, s_2, \ldots, s_n$ be i.i.d. draws from a distribution supported on $[0, \kappa]$. Let $v_i = s_i \cdot x_i$. Define $Z_c = \sum_{i=1}^n \exp(\langle v_i, c\rangle)$, and define $Z_{c,a} = \sum_{i=1}^n \exp(\langle v_i, c\rangle + T(v_a, v_i, c))$.

We first cover the unit sphere by a finite number of metric balls of small radius. Then we show that with high probability, the partition function at the center of these balls is indeed bounded below by a constant. Finally, we show that the partition function evaluated at an arbitrary point on the unit sphere can't be too far from the partition function at one of the ball centers provided the norms of the $v_i$ are not too large. We finish by appropriately controlling the norms of the $v_i$.

For $\epsilon > d^{-1}$, cover the unit sphere in $\mathbb{R}^d$ with $N = (\frac{2}{\epsilon} + 1)^d$ balls of radius $\epsilon$. Let $c_1, c_2, \ldots, c_N$ be the centers of these balls (so that each $c_i$ is a unit vector). Let $\alpha \geq 0$ be a constant. Note that $\langle v_j, c_i\rangle = \langle c_j, s_j \cdot c_i\rangle$ and $\langle v_k, c_i\rangle + T(v_l, v_k, c_i) = \langle x_k, s_k(I + T(v_l, \cdot, \cdot))^T c_i\rangle$ are Gaussian random variables with mean 0.

Let $\mathcal{F}_i$ be the event that there exists some $j, k \in [n]$ such that $\langle v_j, c_i\rangle \geq 0$ and $\langle v_k + T(v_a, v_k, \cdot), c_i\rangle \geq 0$. Note that

$$\begin{aligned}
\Pr[\bar{\mathcal{F}}_i] &\leq \Pr[\forall j \in [n], \langle v_j, c_i\rangle \leq 0] + \Pr[\forall k \in [n], \langle v_k + T(v_a, v_k, \cdot), c_i\rangle \leq 0] \\
&= \prod_{j=1}^n \Pr[\langle v_j, c_i\rangle \leq 0] + \prod_k^n \Pr[\langle v_k + T(v_a, v_k, \cdot), c_i\rangle \leq 0] \\
&\leq \frac{1}{2^n} + \frac{1}{2^n} \\
&\leq \exp(-\Theta(n)).
\end{aligned}$$

Let $\gamma > 0$. Let $\mathcal{G}_i$ be the event that $y_i < \gamma\sqrt{d}$. Set $t = (\frac{1}{\sqrt{2}}\sqrt{\gamma^2 - \frac{1}{2}} - \frac{1}{2})^2 d$, so that $d + 2\sqrt{dt} + 2t = \gamma^2 d$. Then by Lemma 7,

$$\Pr[\bar{\mathcal{G}}_i] \leq \exp(-t).$$

Let $\mathcal{E} = \bigcap_{i=1}^N \mathcal{F}_i \bigcap_{i=1}^n \mathcal{G}_i$. Assume that the word embeddings satisfy the event $\mathcal{E}$. Let $c_i$ be a center of one of the covering balls such that $\|c - c_i\|_2 < \epsilon$. Let $v_j, v_k$ be vectors that satisfies $\langle x_j, c_i\rangle \geq -\alpha$ and $\langle v_k + T(v_a, v_k, \cdot), c_i\rangle \geq -\alpha$. By Cauchy-Schwarz and the definition of $\mathcal{E}$, we have

$$\begin{aligned}
\langle v_j, c\rangle &= \langle v_j, c_i\rangle + \langle v_j, c - c_i\rangle \\
&\geq -\|v_j\|\|c - c_i\| \\
&\geq -\epsilon\gamma\kappa\sqrt{d} \\
&= -\gamma\kappa d^{-1/2} \\
&\geq \ell
\end{aligned}$$

for some appropriate universal constant $\ell$. Likewise, using the boundedness property of $T$, we have

$$\begin{aligned}
\langle v_k + T(v_a, v_k, \cdot), c\rangle &\geq -\epsilon\sqrt{K}\sqrt{d} \\
&= -\sqrt{K}d^{-1/2} \\
&\geq \ell.
\end{aligned}$$

Hence,

$$Z_c = \sum_{i=1}^{n} \exp(\langle v_i, c \rangle) \geq \exp(\langle v_j, c \rangle) \geq \exp(-\ell)$$

and

$$Z_{c,a} = \sum_{i=1}^{n} \exp(\langle v_i, c \rangle + T(v_a, v_i, c)) \geq \exp(\langle v_k + T(v_a, v_k, \cdot), c \rangle) \geq \exp(-\ell).$$

It remains to analyze the probability of $\mathcal{E}$. By the union bound, we have

$$\Pr[\mathcal{E}] \geq 1 - N \exp\left(-\frac{n\alpha^2}{2}\right) - n \exp(-t)$$

$$= 1 - \exp(O(d \log d) - \Theta(n)) - \exp(\log n - (\frac{1}{\sqrt{2}}\sqrt{\gamma^2 - \frac{1}{2}} - \frac{1}{2})^2 d)$$

$$= 1 - \exp(\Theta(d \log d) - \Theta(n)) - \exp(\Theta(\log n) - \Theta(d)).$$

Note that this is a high probability if $n \gg d \log d$ and $d \gg \log n$. $\qquad\square$

