# OpenReview forum: "Understanding Composition of Word Embeddings via Tensor Decomposition"
_ICLR.cc/2019/Conference_

### Official Review · AnonReviewer3 · 2018-11-01
**The paper aims to produce useful word embedding compositions using a method based on the Tucker decomposition of a three-way PMI tensor. The paper presents a potentially promising solution to the problem of compositions in word embedding; yet it is marred by lack of theoretical insights, unwarranted over-generalizations, leaps in justification, and sub-optimal presentation.**

**Rating:** 6
**Confidence:** 3

**Review:**



The authors suggest a method to create combined low-dimensional representations for combinations of pairs of words which have a specific syntactic relationship (e.g. adjective - noun). Building on the generative word embedding model provided by Arora et al. (2015), their solution uses the core tensor from the Tucker decomposition of a 3-way PMI tensor to generate an additive term, used in the composition of two word embedding vectors.

Although the method the authors suggest is a plausible way to explicitly model the relationship between syntactic pairs and to create a combined embedding for them, their presentation does not make this obvious and it takes effort to reach the conclusion above. Unlike Arora's original work, the assumptions they make on their subject material are not supported enough, as in their lack of explanation of why linear addition of two word embeddings should be a bad idea for composition of the embedding vectors of two syntactically related words, and why the corrective term produced by their method makes this a good idea. Though the title promises a contribution to an understanding of word embedding compositions in general, they barely expound on the broader implications of their idea in representing elements of language through vectors.

Their lack of willingness to ground their claims or decisions is even more apparent in two other cases. The authors claim that the Arora's RAND-WALK model does not capture any syntactic information. This is not true. The results presented by Arora et al. indeed show that RAND-WALK captures syntactic information, albeit to a lesser extent than other popular methods for word embedding (Table 1, Arora et al. 2015). Another unjustified choice by the authors is their choice of weighing the Tensor term (when it is being added to two base embedding vectors) in the phrase similarity experiment. The reason the authors provide for weighing the composition Tensor is the fact that in the unweighted version their model produced a worse performance than the additive composition. One would at least expect an after-the-fact interpretation for the weighted tensor term and what this implies with regard to their method and syntactic embedding compositions in general.

Arora's generative model for word embeddings, on which the current paper is largely based upon, not only make the mathematical relationship among different popular word embedding methods explicit, but also by making and verifying explicit assumptions with regard to properties of the word embeddings created by their model, they are able to explain why low-dimensional embeddings provide superior performance in tasks that implicate semantic relationships as linear algebraic relations. Present work, however interesting with regard to its potential implications, strays away from providing such theoretical insights and suffices with demonstrating limited improvements in empirical tasks.

---

> ### Author Response · Authors · 2018-11-19
> **Response to AnonReviewer3**
>
> We thank the reviewer for reading and evaluating our submission.
>
> Additive composition vs. tensor: as discussed in our introduction (and illustrated by the qualitative results in Tables 1 and 2), we believe that linear addition of two word embeddings may be an insufficient representation of the phrase when the combined meaning of the words differs from the individual meanings. Syntactically related word pairs such as adjective-noun and verb-object pairs can have this property. The tensor term can capture the specific meaning of the word pair taken as a whole, as evidenced by qualitative and quantitative evaluations.
>
> RAND-WALK and syntax: we will clarify this point more carefully: what we mean is that the RAND-WALK model itself does not treat syntactically related word-pairs different from other word pairs. From a purely model perspective, in the RAND-WALK model each word is generated independent of all others given the discourse vector, hence the model itself does not account for syntactic relationships between words. Certainly the word embeddings trained based on this model may capture syntactic information that is communicated through the co-occurrence statistics of the training corpus, which allows their embeddings to perform decently on syntactic analogy tasks. Our goal is to explicitly model syntactic dependencies in the context of a word embedding model, in the hopes that the learned embeddings might capture additional information that is missed in non-syntax-aware embedding models.
>
> Weighting the tensor term: we don't expect that our model or any other model will correspond perfectly with how humans use language in practice. When it comes to tasks such as predicting phrase similarity, we give our model a bit of extra flexibility to account for this discrepancy. We also note that previous works on embedding composition also explore various re-weighting schemes. While the meaning of the weighting parameter isn't a central question in our work, one can think of it as the degree to which specific knowledge of the syntactic relationship between the two words affects the phrase's overall meaning.
>
> Verifying assumptions in our model: we note that in section 5 of the paper, we verify the assumptions and concentration phenomena introduced in our model.

---

### Official Review · AnonReviewer1 · 2018-11-02
**novel, but it is unclear that the approach is useful for downstream tasks**

**Rating:** 6
**Confidence:** 4

**Review:**

The authors consider the use of tensor approximations to more accurately capture syntactical aspects of compositionality for word embeddings. Given two words a and b, when your goal is to find a word whose meaning is roughly that of the phrase (a,b), a standard approach to to find the word whose embedding is close to the sum of the embeddings, a + b. The authors point out that others have observed that this form of compositionality does not leverage any information on the syntax of the pair (a,b), and the propose using a tensor contraction to model an additional multiplicative interaction between a and b, so they propose finding the word whose embedding is closest to a + b + T*a*b, where T is a tensor, and T*a*b denotes the vector obtained by contracting a and b with T. They test this idea specifically on the use-case where (a,b) is an adjective,noun pair, and show that their form of compositionality outperforms weighted versions of additive compositionality in terms of spearman and pearson correlation with human judgements. In their model, the word embeddings are learned separately, then the tensor T is learned by minimizing an objective whose goal is to minimize the error in predicting observed trigram statistics. The specific objective comes from a nontrivial tensorial extension of the original matricial RAND-WALK model for learning word embeddings.

The topic is fitting with ICLR, and some attendees will find the results interesting. As in the original RAND-WALK paper, the theory is interesting, but not the main attraction, as it relies on strong generative modeling assumptions that essentially bake in the desired results. The main appeal is the idea of using T to model syntactic interactions, and the algorithm for learning T. Given that the main attraction of the paper is the potential for more performant word embeddings, I do not believe the work will have wide appeal to ICLR attendees, because no evidence is provided that the features from the learned tensor, say [a, b, T*a*b], are more useful in downstream applications than [a,b] (one experiment in sentiment analysis is tried in the supplementary material with no compelling difference shown).

Pros:
- theoretical justification is given for their assumption that the higher-order interactions can be modeled by a tensor
- the tensor model does deliver some improvement over linear composition on noun-adjective pairs when measured against human judgement

Cons:
- no downstream applications are given which show that these higher order interactions can be useful for downstream tasks.
- the higher-order features T*a*b are useful only when a is noun and b is an adjective: why not investigate using T to model higher-order interaction for all (a,b) pairs regardless of the syntactic relationships between a and b?
- comparison should be made to the linear composition method in the Arora, Liang, Ma ICLR 2017 paper

Some additional citations:
- the above-mentioned ICLR paper provides a performant alternative to unweighted linear composition
- the 2017 Gittens, Achlioptas, Drineas ACL paper provides theory on the linear composition of some word embeddings

---

> ### Author Response · Authors · 2018-11-19
> **Response to AnonReviewer1**
>
> We are grateful to the reviewer for their time and effort in reading our paper and providing feedback.
>
> Generative model assumptions: our model is an expansion of the original RAND-WALK model of Arora et. al., with the purpose of accounting for syntactic dependencies. The additional assumptions we include and the concentration phenomena we prove theoretically are verified empirically in section 5, so our results do hold up on real data.
>
> Use on downstream tasks: we believe that capturing syntactic relationships using a tensor can be useful for some downstream tasks, since our results in the paper suggest that it captures additional information above and beyond the standard additive composition. However, as the main goal of this paper is to introduce and analyze the model, we defer more application-focused analysis to future work.
>
> Interaction between arbitrary word pairs: our model introduces the tensor in order to capture syntactic relationships between pairs of words, such as adjective-noun and verb-object pairs. While it might be interesting to try to capture interactions between all pairs of words, that is not justified by our model and we didn't explore it. However, we also trained our model using verb-object pairs, and we have updated section 5 as well as the appendix to include these additional results.
>
> Comparison to Arora, Liang, Ma ICLR 2017: we appreciate the suggestion to include a comparison with the SIF embedding method of Arora et. al., as this method is also obtained from a variant of the original RAND-WALK paper. We have updated Table 2 and the discussion in section 5 to include these additional results. As reported in their paper, the SIF embeddings yield a strong baseline for sentence embedding tasks, and we find the same to be true in the phrase similarity task for adjective-noun phrases (not so for verb-object phrases). However, we find that we can improve upon the SIF performance by addition of the tensor component from our model. (We note that we have just used the tensors trained in our original model; it is possible that combining the model in SIF and syntactic RAND-WALK more carefully could give even better results.)
>
> Additional citations: we have updated the paper to include both additional citations.

---

### Official Review · AnonReviewer2 · 2018-11-03
**T(v_a, v_b,.)-addition is an improvement?**

**Rating:** 7
**Confidence:** 2

**Review:**

The paper deals with further development of RAND-WALK model of Arora et al. There are stable idioms, adjective-noun pairs and etc that are not covered by RAND-WALK, because sometimes words from seemingly different contexts can join to form a stable idiom.

So, the idea of paper is to introduce a tensor T and a stable idiom (a,b) is embedded into v_{ab}=v_a+v_b+T(v_a, v_b,.) and is emitted with some probability p_sym (proportional to exp(v_{ab} times context)). The latter model is similar to RAND-WALK, so it is not surprising that statistical functions there are similarly concentrated. Finally, there exists an expression, PMI3(u,v,w), that shows the correlation between 3 words, and that can be estimated from the data directly. It is proved that Tucker decomposition of that tensor gives us all words embeddings together with tensor T. Thus, from the latter we will obtain a tool for finding embeddings of idioms (i.e. v_a+v_b+T(v_a, v_b,.)).

Theoretical analysis seems correct (I have not checked all the statements thoroughly, but I would expect formulations to be true). The only problem I see is that phrase similarity part is not convincing. I cannot understand from that part whether T(v_a, v_b,.) addition to v_a+v_b gives any improvement or not.

---

> ### Author Response · Authors · 2018-11-19
> **Response to AnonReviewer2**
>
> We thank the reviewer for their time and response to our paper.
>
> Phrase similarity results: the tensor component T(v_a,v_b,.) does yield improvement over all other weighted additive methods in 5 out of 6 cases, as shown in Table 3. We have also updated that table with additional results, which show that adding in the tensor component improves upon the strong baseline of the SIF embedding method. We also added Table 4, which repeats the phrase-similarity task for verb-object pairs, and shows that the tensor component leads to improvement in most cases.

---

### Author Response · Authors · 2018-11-19
**Uploaded revision**

We have uploaded a revision of the paper that incorporates suggestions of the reviewers and expands on experimental results. The largest changes are in Section 5 on the experimental verification, where we include the results of our experiments on verb-object phrases (previously we only showed results for adjective-noun phrases).

---

### Meta-Review · Area_Chair1 · 2018-12-15
**Further development/continuation of Arora et al.**

**Confidence:** 4
**Recommendation:** Accept (Poster)

**Metareview:**

AR1 is concerned about lack of downstream applications which show that higher-order interactions are useful and asks why not to model higher-order interactions for all (a,b) pairs. AR2 notes that this submission is a further development of Arora et al. and is satisfied with the paper. AR3 is the most critical regarding lack of explanations, e.g. why linear addition of two word embeddings is bad and why the corrective term proposed here is a good idea. The authors suggest that linear addition is insufficient when final meaning differs from the individual meanings and show tome quantitative results to back up their corrective term.

On balance, all reviewers find the theoretical contributions sufficient which warrants an accept. The authors are asked to honestly reflect all uncertain aspects of their work in the final draft to reflect legitimate concerns of reviewers.